# Influencing Factors of Direct Carbon Emissions of Households in Urban Villages in Guangzhou, China

**DOI:** 10.3390/ijerph192417054

**Published:** 2022-12-19

**Authors:** Yamei Chen, Lu Jiang

**Affiliations:** 1School of Geography Science, Qinghai Normal University, Xining 810008, China; 2Faculty of Geographical Science, Beijing Normal University, Beijing 100875, China

**Keywords:** urban village, household carbon emissions, influencing factors, questionnaire

## Abstract

China’s household energy consumption has obvious regional differences, and rising income levels and urbanization have changed the ability of households to make energy consumption choices. In this paper, we analyze the energy consumption characteristics of urban village residents based on microlevel household survey data from urban villages in Guangzhou, China. Then, the results of modeling the material flows of per capita carbon emissions show the most dominant type of energy consumption. OLS is applied to analyze the influencing factors of carbon emissions. We find that the per capita household carbon emissions in urban villages are 722.7 kg/household.year, and the average household carbon emissions are 2820.57 kg/household.year. We also find that household characteristics, household size, household appliance numbers, and carbon emissions have a significant positive correlation, while income has no significant effect on carbon emissions. What is more, the size and age of the house have a positive impact on carbon emissions. Otherwise, the new finding is the demonstration that income is not significantly correlated with household carbon emissions, which is consistent with the characteristics of urban villages described earlier. On the basis of this study, we propose more specific recommendations regarding household energy carbon emissions in urban villages.

## 1. Introduction

Urbanization is changing the global environment at an unprecedented speed and scale and has brought great challenges to energy intensity and CO_2_ reduction. Household energy consumption is China’s second-largest energy consumption sector, approaching the industrial sector. It accounts for 14.6% of the country’s total energy consumption and has become an important growth point for China’s energy consumption and carbon emissions [1]. China formally proposed that carbon peaking be achieved by 2030 and carbon neutrality by 2060 at the 75th United Nations General Assembly. “Double carbon” is proposed as a goal [2]. Researchers are paying increasing attention to the relevance of carbon emissions and household elements, and they have proposed many macro and micro policies to achieve carbon emission reduction [3,4,5]. However, China’s long-term dual structure has caused urban residents to have different production and lifestyles than rural residents, and their energy consumption and carbon dioxide emissions characteristics also differ [6]. Therefore, the study of household carbon emissions in urban villages is conducive to the implementation of sustainable consumption patterns and is of great significance to coordinating the development of humankind and cities. Reducing carbon emissions and promoting the construction of low-carbon communities are important ways to promote the development of low-carbon communities and mitigate global warming. As a result of the literature available, it has been difficult to identify a model from the social development of other countries that can be used for reference. The purpose of the paper is to supplement the theoretical deficiencies of these studies. This paper focuses on the energy consumption behavior, characteristics, and energy-saving potential of this group of people.

Urban villages refer to regional entities with an urban–rural dual structure, in which original rural settlements are surrounded by construction land and incorporated into cities due to the rapid development of urban suburbanization, industrial decentralization, and rural urbanization. They represent a product of the urbanization process. The “urban village” has evolved into a “low-income community that provides low-cost housing for the urban transient population” [7]. Many researchers have considered urban villages from the perspective of urbanization and social issues. For example, Qian believes that urban villages fetter the urbanization process [8]. Gao believes that urban villages help immigrants integrate into urban society by providing affordable housing and convenient facilities. However, researchers have paid little attention to the energy consumption and carbon emissions of urban village households [9]. Due to a lack of microdata, it is difficult to accurately identify the energy consumption characteristics and existing problems of urban village households.

Researchers had already carried out many studies in the field of household energy consumption. At the level of applied statistics data analysis, Zhang explored the factors influencing energy consumption and direct and indirect CO_2_ emissions that were related to household consumption using the input–output method. The results showed that direct energy consumption and CO_2_ emissions were the main components of total energy consumption and CO_2_ emissions from household consumption, and indirect CO_2_ emissions showed an increasing trend [10]. Jiang used the Divisia index model to analyze the drivers of carbon emissions and changes in emission intensity. The results showed that from 1996~2012, there was a significant shift in urban residential energy consumption from primary energy consumption to electricity and heat consumption, and direct CO_2_ emissions decreased as a proportion of residential emissions. Second, residential energy intensity, housing area per capita, and number of households were the main drivers of changes in carbon emissions [11]. Du used a cointegration econometric model to analyze how China’s economic growth potential is affected by macroeconomic variables in the context of urbanization and to forecast China’s economic growth potential up to 2020 under different scenarios. The results of the study showed that lifestyle, standard of living and energy prices had significant effects on CO_2_ emissions [12]. Wang used an input–output model to calculate the direct and indirect carbon emissions of urban and rural residents’ consumption in the BTH region from 2002 to 2012. The results showed that the direct and indirect carbon emissions of residents’ consumption increased year by year [13]. Based on multiple regression models, Li conducted urban–rural studies on direct and indirect household carbon emissions across the country. The results showed that household income and the household labor force have important impacts on direct carbon emissions; furthermore, subjective factors had a meaningful impact on the household carbon emissions of urban and rural residents. Moreover, the emission reduction effects of some subjective factors were greater in rural households than in urban households; urban and eastern residents have higher consumption, generating more emissions, and their willingness to reduce emissions is lower, as estimated [14]. At the level of applied micro survey data analysis, Yang used household surveys, by the residential carbon emissions and private transportation carbon emissions of 826 urban households in Beijing. The results showed that the properties of energy-saving buildings and better community facilities had significantly reduced the carbon emission levels of households. Residents of communities far away from public facilities tended to have a higher probability of buying cars, leading to higher traffic carbon emissions [15]. Gu found the CO_2_ emission status of the energy consumption of urban households in Nanjing and its influencing factors were analyzed. The households studied were located in three urban areas of Nanjing. Information on building characteristics, household characteristics, household appliance use, and fuel consumption was obtained through a questionnaire survey. The study found that energy consumption was generally positively correlated with income. Household structure also influences energy consumption, with higher income households and smaller households consuming more energy per capita [16]. Using a stratified random sampling method and applying descriptive statistical analysis and ordinary least squares (OLS) regression to analyze survey data, Xu found that residence area has a significant impact on household energy consumption and carbon emissions [17].

In general, researchers who study carbon emissions from household energy consumption fall into the following categories. In terms of research, there is direct and indirect energy consumption. For area characteristics, there are urban and rural areas. For data use, both survey data and statistical data are available. In some studies, the influencing factors are examined. It is primarily influenced by socioeconomic and geographical factors. In previous studies, household income, age, household size, education level, location, gender, and building time have been shown to have the greatest impact on carbon emissions. Nevertheless, few researchers have examined household carbon emissions from the perspective of urban communities [18].

In India, Nair studied the correlation between household carbon emissions and income and household size in three Indian cities using ANOVA and linear regression models to show that household size and CO_2_ emissions from different activities had a large impact on total emissions [19]. Khosla discussed the changing demand for energy services among low-income households in urban India and showed that on the energy demand side, appliance penetration rises as household consumption capacity increases. In terms of energy behavior, households experience indoor heat differently and use different types of energy for cooling strategies [20]. Baul used a semistructured questionnaire to conduct an exploratory survey of 189 households in three income groups in the suburbs of Chittagong, Bangladesh. They found that low-income families tended to choose traditional biomass energy, and high-income families tended to choose nonrenewable energy [21]. In fact, Guangzhou, India, and Bangladesh are at roughly the same latitude, have similar climates, and are developing regions, but the different levels of urbanization and economic development contribute to different types of household energy consumption. While India and Bangladesh have slums, urban villages are very “Chinese”, and their residents are mostly migrant workers. Hence, the diversity of livelihoods may lead to differences in the types of energy consumption and uses, so there are both similarities and differences among the three [22,23].

In fact, Guangzhou, India, and Bangladesh have certain similarities in terms of climatic characteristics, but the different levels of urbanization and economic development have led to different types of household energy consumption [24]. Urban villages are very much “Chinese in character”, with rapid urbanization driving regional economic growth and leading to the emergence of informal urban living spaces. To focus on such areas, this paper uses microscale questionnaire survey data to conduct direct carbon emissions research on households in typical urban villages and analyzes the energy consumption structure and end uses leading to carbon emissions. Multiple regression is used to study the influencing factors of household carbon emissions in urban villages, and targeted recommendations are proposed for the low-carbon, sustainable development of urban villages. The theoretical contribution of the article is that it pays attention to the energy consumption behavior and characteristics of this group of people and promotes their energy-saving potential, but it is difficult to find a model that can be used as a reference from the social development of other countries [25].

## 2. Data and Methodology

### 2.1. Research Area

Guangzhou is located between 112°57′ and 114°3′ east longitude and 22°26′ and 23°56′ north latitude. It has six months of summer per year and a subtropical climate [26]. Guangzhou is a central city in southern China and has a large concentration of urban villages, mainly in the Huangpu, Tianhe, Liwan, Haizhu, and Baiyun districts. The results of the seventh census show that there are new trends in population movement [27]. Areas with large population inflows can experience increasing conflicts between energy supply and demand. The 10 cities with the fastest growing populations in China in 2020 were Shenzhen, Chengdu, Guangzhou, Zhengzhou, Xi’an, Hangzhou, Chongqing, Changsha, Wuhan, and Foshan, with a total population of 158 million, 42.1 million more than in 2010. Their share of the national population is projected to increase from 8.7% in 2010 to 11.2% in 2020. Figure 1 shows the location of Guangzhou and Guangdong Province in China.

### 2.2. Data Sources and Sample Characteristics

A total of 247 valid samples were collected using field research sampling and in-depth interviews in urban villages in Guangzhou in 2020. Stratified random sampling was used. The survey included basic household information, kitchen equipment and household appliances, electricity, transportation, and housing consumption. The average age of the respondents was 32.5, with the youngest being 18 years old and the oldest being 60 years old. In terms of gender, 47.77% were female and 52.23% were male. In terms of occupation, most of the respondents were office workers, general workers, and freelancers, accounting for 62.34%, and the average education level of the respondents was tertiary level. The chart below shows the structure of the questionnaire design (Figure 2) [28].

### 2.3. Calculation and Analysis of Household Energy Carbon Emissions

#### 2.3.1. Household Carbon Emissions Calculation

The main types of energy consumed by urban village households in Guangzhou city are electricity, liquefied petroleum gas and natural gas. This study is based on the accounting methods provided in the IPCC Guidelines for Greenhouse Gas Inventories and uses the following calculation methods [29]:

(1) The formula for carbon emissions from household electricity is:(1)$$C_{e}=E_{e}\times EF_{e}$$
where $C_{e}$ is the CO_2_ emissions from household electricity (kg), $E_{e}$ is the household electricity (kWh), and $EF_{e}$ is the carbon emissions factor of the power grid (tCO_2_/MWh).

(2) The formula for carbon emissions from household natural gas is:(2)$$C_{n}=E_{n}\times EF_{n}\times NV_{n}\times 44/12$$
where $C_{n}$ is the CO_2_ emissions from natural gas (kg), $E_{n}$ is the domestic natural gas volume (m^3^), $EF_{n}$ is the natural gas carbon emissions coefficient (kg/J), and $NV_{n}$ is the average low calorific value (J).

(3) The formula for carbon emissions from household liquefied petroleum gas is:(3)$$C_{{}_{l}}=E_{{}_{l}}\times EF_{{}_{l}}\times NV_{{}_{{}_{l}}}\times 44/12$$
where $C_{l}$ is the domestic CO_2_ emissions of liquefied petroleum gas (kg), $E_{{}_{l}}$ is the liquefied petroleum gas volume (m^3^), $EF_{{}_{l}}$ is the carbon emissions coefficient of liquefied petroleum gas (kg/J), and $NV_{{}_{{}_{l}}}$ is the average low calorific value (J).

(4) The formula for total carbon emissions from household energy consumption is:(4)$$CH=C_{e}+C_{n}+C_{l}$$
where $C_{n}$ is the total carbon emissions from household energy consumption (kg), $C_{e}$ is the CO_2_ emissions from household electricity (kg), $C_{n}$ is the CO_2_ emissions from natural gas (kg), and $C_{l}$ is the domestic CO_2_ emissions of liquefied petroleum gas (kg).

#### 2.3.2. Basic Information on Household Carbon Emissions

The main types of energy used by residents in urban villages in Guangzhou are electricity, LPG, natural gas, and others. Of the 247 households surveyed, 44 households used natural gas for cooking, 145 households used tank gas, 52 households used electricity, and 6 households used other fuels. In terms of average household carbon emissions, electricity emissions averaged 1352.99 kg/household.year, and LPG and natural gas emissions averaged 1467.56 kg/household.year. The total direct household carbon emissions were 2820.55 kg/household.year. Electricity, LPG, and natural gas accounted for 47.97%, 36.1%, and 15.93% of direct carbon emissions, respectively, with electricity consumption being the main household energy carbon emission. In terms of annual per capita carbon emissions, electricity emits 346.67 kg > LPG 260.89 kg > natural gas 115.14 kg. Figure 3 shows the separate CO_2_ emissions.

#### 2.3.3. Comparative Analysis with Other Cities in China

A first comparison with the traditional community in Beijing, the capital of China, shows annual carbon emissions of 4732.8 kg/household.year, which can be explained by two main factors. On the one hand, the energy mix of the traditional community in Beijing is richer than that of the urban village in Guangzhou, with two more energy sources, namely coal and heat required for heating. On the other hand, the people living in this community, although traditional, are characterized by a high level of environmental satisfaction and involvement in scientific research. Those involved in scientific research are generally more educated and have more demands regarding quality of life. The characteristics of this group structure differ from the characteristics of the inhabitants of traditional communities, who are generally older, prefer a simpler life, and concentrate their energy consumption on basic daily consumption activities; these additional energy sources, however, significantly stimulate the rise in carbon emissions of traditional communities. In contrast, the average age of the population in Guangzhou’s urban villages is 33 years old, and the group structure is characterized by low-income migrant workers with low demands on quality of life; thus, carbon emissions tend to be relatively low [30].

A second comparison with the annual per capita carbon emissions of the old city of Nanjing, a coastal area in southeast China, shows emissions of 1200 kg of carbon compared to 722.7 kg per capita in the urban villages of Guangzhou. Even though the energy mix of the old city of Nanjing is more homogeneous than that of the urban villages of Guangzhou, with only electricity and natural gas, the carbon emissions are still higher. The main reason for this is that in terms of climatic factors, Nanjing is a city with four distinct seasons, which presents a challenge in terms of increasing energy consumption for cooling and heating. Guangzhou has an average annual temperature of 22.7 degrees Celsius and does not require heating in the winter months. In terms of economic income, there is also a large difference. The residents of Guangzhou’s urban villages are low-income migrant workers and are at the lowest level of Maslow’s needs hierarchy, while the old city of Nanjing, although old in terms of construction time, is the closest area to the city center and was the first residential neighborhood to develop. Therefore, there is a large difference in income. The studied inhabitants of Nanjing have a more comfortable lifestyle, and their carbon emissions from electricity consumption alone are 3.3 times higher than those of the urban village residents of Guangzhou [16].

The next comparison with the annua per capita direct household energy emissions in urban areas of Northwest China reveals that the per capita household carbon emissions in Northwest China are higher than those in the urban villages of Guangzhou, reaching 1100 kg. In Guangzhou’s urban villages, only electricity, liquefied petroleum gas and natural gas are used as energy sources. The difference in the calculated carbon emissions in the urban villages of Guangzhou may also be due to the difference in statistical calibrations, as carbon emissions from transportation are not taken into account in the calculation process. Finally, the long and cold winters in the northwest require more energy for heating [31].

In a final comparison with the central city of Kaifeng, the city’s annual per capita carbon emissions are 1073.1 kg, higher than the per capita carbon emissions of the urban village of Guangzhou. This discrepancy is because Kaifeng has a richer energy mix than Guangzhou, and electricity accounts for 75.28%, 27.31% higher consumption than in Guangzhou’s urban villages. Additionally, central heating in Kaifeng accounts for 9.06%, while Guangzhou has no central heating, hence the difference in carbon emissions [32].

Overall, the direct carbon emissions of the households in the cities compared are higher than those of the urban village households in Guangzhou. The reasons for this are twofold. In terms of climatic factors, Guangzhou has a subtropical monsoon climate and does not require heating in winter, while the comparison cities all require heating in winter. In terms of economic income, Guangzhou’s urban villages are inhabited by migrant workers who do not suffer from energy poverty but use energy to meet their basic needs [33]. As shown in Figure 4.

### 2.4. Modeling of Carbon Emissions Energy Flows per Capita

To further analyze household carbon emission activities in detail, a typical per capita household energy carbon emission material flow model was developed to account for household energy access, structure, use, and pollutant emissions and to estimate the types of emissions produced, pollutant emission factors, etc., based on the combustion of each energy source. Power BI (Sankey 3.0.3) software was applied to map the energy material flows (Figure 5) [34].

First, it is known that residents of urban villages use commodity energy. A calculation of the carbon emissions per capita of household energy shows that the carbon emissions per capita of electricity are 346.67 kg, mainly for cooking, refrigeration, and home appliances. The per capita carbon emissions from cooking were 37.21 kg, from household appliances, 95.95 kg and from refrigeration, 231.51 kg, accounting for 61.59% of the total carbon emissions from electricity, which shows that air conditioning and refrigeration are the main causes of carbon emissions from electricity. In contrast, LPG produces 260.89 kg of carbon emissions per capita and is mainly used for cooking. Natural gas produces 115.14 kg of carbon emissions per capita and is also mainly used for cooking. LPG is an important source of cooking energy, accounting for 63.13% of cooking carbon emissions. Finally, from the perspective of the end consumer, cooking produces the most air pollutants because it employs a relatively diverse energy mix used for cooking. Constructing the household per capita carbon emissions energy flow offers a clearer understanding of the structure and use of household per capita carbon emissions.

## 3. Results

### 3.1. Variable Selection and Description

This article uses a multiple regression model to perform an explanatory analysis of household energy carbon emissions. First, a preliminary understanding of each variable is gained with descriptive statistics, and the average value, standard deviation, minimum value, and maximum value of each variable are understood. As shown in Table 1.

### 3.2. OLS Regression Model

This article establishes the classic least squares model for regression. For the key variables in this step, stepwise regression is used to screen for insignificant variables. The variance inflation factor (VIF) is less than 10, and the DW test is 2.5, indicating that there is no collinearity or autocorrelation between the variables.

Taking into account the interpretability of the variables, the permanent household population, family annual income, housing area, and building age are finally selected as core explanatory variables, and the numbers of air conditioners and refrigerators in the family are used as control variables to perform regression analysis on the total carbon emissions of the family [35].

The formula is as follows:(5)$$LNY_{i}=\beta_{0}X_{0}+\beta_{1}X_{1}+\beta_{2}X_{2}+\beta_{3}X_{3}+\beta_{4}X_{4}+\beta_{5}X_{5}+\beta_{6}X_{6}+\varepsilon$$

In Formula (5), $LNY_{i}$ is the dependent variable, which represents household energy carbon emissions; β is the regression coefficient; X is the independent variable, which represents the factors affecting household carbon emissions; and εis  is the random error term of the model. The regression results are as follows Table 2 [36]:

### 3.3. Regression Analysis

According to the regression results, the permanent household population, housing area, building age, number of air conditioners, and number of refrigerators have significant effects on total household carbon emissions to varying degrees. Except for annual household income, other influencing factors have a significant impact on total household carbon emissions. The permanent household population, housing area, building age, number of refrigerators, and number of air conditioners have a significant positive impact on the total carbon emissions of the household. The t test and *p* value show that the indicators selected in this paper are reasonable.

First, the regression coefficient shows a nonsignificant positive relationship between household income and household carbon emissions, mainly because rental housing in urban villages provides cheap housing for millions of migrant workers. These workers are low-income people who live in urban villages and work in labor-intensive industries and business services in the city. Second, there is a positive correlation between household size and total household carbon emissions at the 95% significance level, meaning that as the number of people living in a household increase, the amount of energy required increases, leading in turn to an increase in total household carbon emissions. Once again, buildings built 20 years ago or more have an average increase in total household carbon emissions of 0.26, which is significant at the 90% level, compared to buildings built less than five years ago. This is mainly because houses built earlier are not considered to be energy-efficient buildings. For example, these houses are poorly insulated and thus require considerable energy to maintain indoor temperatures. Finally, housing size has a positive relationship with total household carbon emissions. Households with a housing area of 70–100 m^2^ and 100 m^2^ or more have an increase in total household carbon emissions of 0.28 compared to those with a housing area of 50 m^2^ or less, which is significant at the 90% level. This indicates that as the size of the house increases, the energy demand for household appliances gradually increases, resulting in an increase in total household carbon emissions.

In terms of control variables, as the number of air conditioners owned by households increases, there is an increase in total household carbon emissions. The average increase in total household carbon emissions for households with one, two and three or more units compared to those without air conditioning is 0.49, 0.84, and 0.97, respectively, significant at the 95% and 99% significance levels. This is mainly due to the year-round hot climate in Guangzhou, with high summer temperatures and a high demand for air conditioning, which in turn leads to an increase in carbon emissions. The number of refrigerators in a household has a positive relationship with total household carbon emissions; emissions in households with one or two refrigerators increase by 0.60 and 0.51 compared to households without refrigerators, and the coefficients are significant at the 99% and 95% significance levels. Refrigerators are a necessary household appliance, and more than 90% of households own at least one; their use increases household carbon emissions to some extent.

## 4. Discussions

To explain the reasons behind these phenomena, we modeled the factors that influence household energy consumption in urban villages in Guangzhou. Comparatively, existing studies have focused more on the temporal variation and urban–rural differences in household carbon emissions from energy consumption [37]. In this study, we examine a unique community property, the urban village, from the perspective of a new perspective. There is no significant effect of household income on carbon emissions in urban villages in Guangzhou, which is inconsistent with other research. We also examined the impact of building time and floor area on carbon emissions in urban villages in the hope of providing some suggestions for creating green, low-carbon and energy-efficient buildings. Our findings provide insight into the direct carbon emissions associated with particular communities and can serve as a basis for future research [38].

(1) The characteristics of households.

There is no significant correlation between household income and carbon emissions. The findings are inconsistent with those of Meangbua and De [39,40]. According to these two researchers, household income has a significant impact on carbon emissions from energy consumption. In spite of this, our study indicates that there is no significant correlation between household income and household energy carbon emissions. We believe that this is one of the most important contributions of our study since it fully explains the residential characteristics of urban villages in Guangzhou, most of whom are low-income migrant workers. The greater the number of household members, the greater the total household carbon emissions, a finding that is consistent with those of other researchers. In view of the fact that people are the primary consumers of energy, this is reasonable. Carbon emissions are also influenced by the number of household appliances owned [41]. Based on Tran and Lee’s argument, it can be shown that the more household appliances used in a household, the greater the impact on carbon emissions [42,43].

(2) The characteristics of housing.

The percentage of buildings in urban villages that are older than 15 years is 52.5%, more than half. A longer building time indicates poor thermal performance, which results in higher electricity consumption. This is consistent with the findings of Yu and Gu’s study [16,44]. There is a significant impact of floor space on carbon emissions in terms of area. It is generally true that the larger the size of a household building is, the more energy is consumed for cooling. Thus, reasonable and effective control of housing size is also an effective method of achieving low carbon development [45].

## 5. Conclusions and Policy Implications

In terms of the theoretical contribution, the conclusion of the paper is conducive to reducing carbon emissions, promoting the construction and development of low-carbon communities, and mitigating global warming. It contributes to the promotion of energy efficiency and emission reduction awareness among consumers, as well as reducing carbon emissions from the consumption side of the equation. What is more, the practical contribution consists of facilitating residents’ participation in the development of low-carbon, sustainable communities. In addition, it is beneficial for building the sustainable community, especially for the lower income households. Thus, the results of our study can provide policymakers with a good case to make policy.

(1) In terms of energy consumption structure and use, household energy consumption is an important part of urban energy consumption, and the resulting carbon emissions are an important constraint on urban development. Many researchers believe that urban villages are an obstacle to sustainable urban development. Our analysis of the results shows that electricity emissions are relatively high in urban villages, mainly due to the ease of access to electricity and the fact that most household appliances require charging to be used; natural gas and LPG are relatively low in proportion and are mainly used for cooking. This situation occurs because of the poor natural gas infrastructure in urban villages; the use of clean energy is very low. Solar energy is used in a very small proportion of households for hot water. The current energy consumption structure is not conducive to low-carbon urban development. Guangzhou’s climatic characteristics and the population and income of its urban villages are the main reasons for the difference in carbon emissions with regard to other cities in China.

(2) From the constructed material flow modeling of carbon emissions per capita, it is clear that urban village residents use all commodity energy. The per capita carbon emissions from refrigeration are 231.51 kg, accounting for 61.59% of the total carbon emissions from electricity. Air conditioning and refrigeration are the main causes of carbon emissions from electricity. LPG is an important source of cooking energy, accounting for 63.13% of cooking carbon emissions.

From the analysis of influencing factors, household characteristics are an important factor influencing direct household carbon emissions. The number of household appliances and the number of people living in the household are the main factors influencing household carbon emissions, and there is no doubt that when the use of household appliances increases, household carbon emissions increase. In terms of residential characteristics, the increase in the size of the dwelling causes an increase in carbon dioxide emissions, with the age of the building having the greatest impact on carbon emissions when the building is over 20 years old. The main reason for this impact is that the buildings in urban villages are old. Therefore, houses are not considered to be energy-efficient buildings.

Based on the above findings, the following policy implications are made: (1) In terms of consumption behavior, based on the characteristics of people living in urban villages and feedback on energy-saving awareness, we should increase efforts to promote low carbon and environmental protection, strengthen publicity on green lifestyles, guide the reasonable control of indoor air conditioning temperatures, and enhance residents’ concepts and awareness of energy-saving consumption. In addition, when purchasing household appliances, residents should choose energy-efficient appliances as much as possible, especially for appliances with large carbon emissions, such as air conditioners, try to turn off household appliances and lighting when not needed, and use efficient lighting products. (2) In terms of the energy application structure, the proportion of natural gas applications should be further increased, and the natural gas infrastructure should be improved. The proportion of clean energy, such as solar, wind, tidal and nuclear power, should be increased so that carbon emissions from the production side of electricity can be reduced. (3) In terms of community residential construction, consideration should be given to whether older buildings are energy-efficient, and energy-efficient materials should be used in subsequent house construction. Houses that are currently occupied can be retrofitted with measures to improve insulation. Residents should be encouraged to actively participate in the governance of the community, as each resident is a member of society and has a responsibility and obligation to contribute to the low carbon, livable and sustainable development of the community. (4) The government should promote new energy trams and implement price concessions and subsidies for purchasing clean energy. Technological means should be used to improve fuel efficiency. Resident travel modes should be optimized to offer multiple options for public transport, thereby reducing carbon emissions from private car travel. The government should improve energy pricing mechanisms, implement tiered tariffs, regulate residents’ energy demand, and use market pricing mechanisms wisely to guide residents toward reasonable consumption. Knowledge of sustainable development should be disseminated so that residents can become aware of and involved in sustainable development and quantify this goal.

The study primarily relies on micro surveys for its data, whose quality and quantity are highly relevant to its credibility. In the future, we need to open up access to more accessible and relevant data to be able to study carbon emissions from household energy consumption more accurately and in depth. Due to limited resources, our study, like many survey studies, requires a larger valid sample to be accurate. In the future, follow-up studies should expand the sample size and examine the management model and sustainable development of urban village communities.

## Figures and Tables

**Figure 1 ijerph-19-17054-f001:**
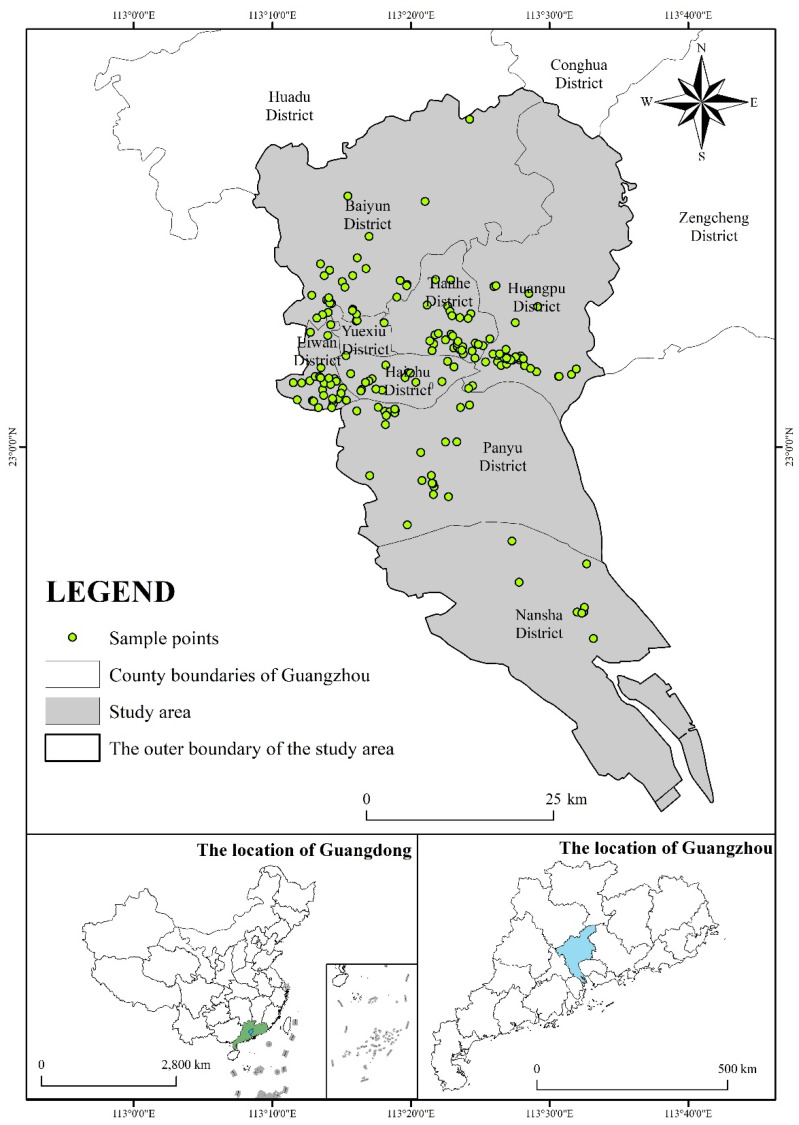
The survey sample in the urban village area of Guangzhou.

**Figure 2 ijerph-19-17054-f002:**
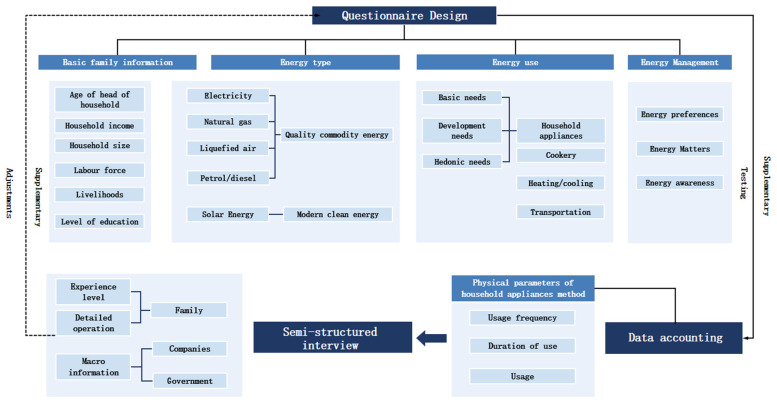
Survey process, methods and data accounting [28].

**Figure 3 ijerph-19-17054-f003:**
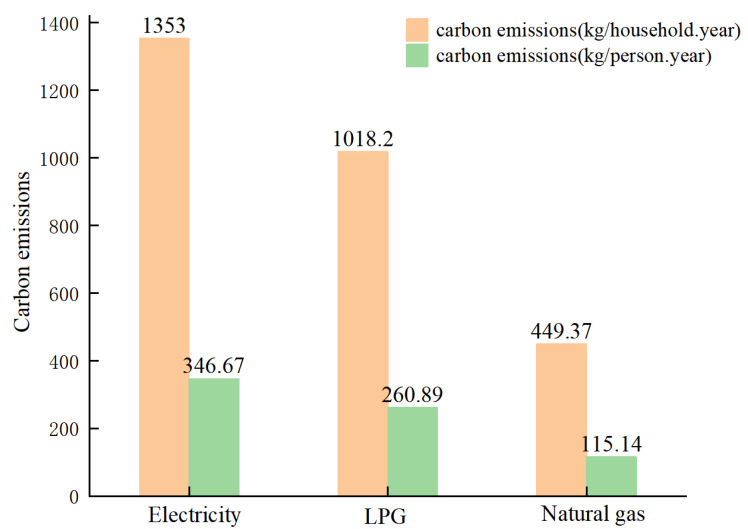
The carbon emissions from energy consumption by type.

**Figure 4 ijerph-19-17054-f004:**
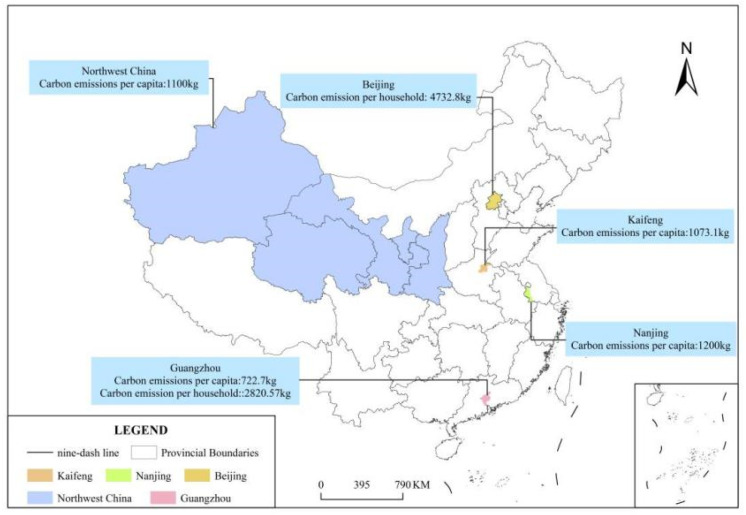
Map of comparison cities.

**Figure 5 ijerph-19-17054-f005:**
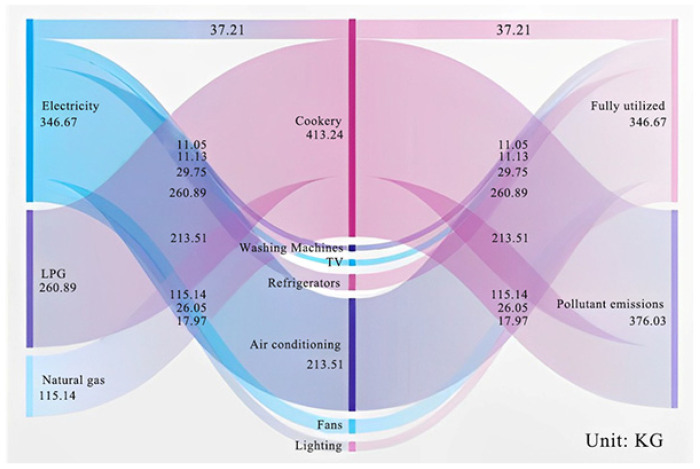
Modeling of material flows of carbon emissions per capita from household energy consumption.

**Table 1 ijerph-19-17054-t001:** Variable descriptive statistical analysis.

Category	Variable	Assignment Description	Mean	Standard Deviation	Minimum	Max
Explained variable	Ln household carbon emissions	Take the logarithm of total household carbon emissions	7.68	0.81	4.66	9.61
Explanatory variables	Family income	1 = Below 50,000 yuan 2 = 5–10 million 3 = 100,000–15 million 4 = 15–20 million 5 = 20–30 million 6 = More than 300,000	2.95	1.57	1	6
	Building age	1 = Less than 5 years 2 = 5–10 years 3 = 10–20 years 4 = More than 20 years	2.51	0.94	1	4
	Construction area	1 = Below 50 square meters 2 = 50–70 square meters 3 = 70–100 square meters 4 = 100 square meters or more	2.37	1.05	1	4
	Permanent household population	1 = 1 person 2 = 2 people 3 = 3 people 4 = 4 people 5 = 5 people 6 = 6 people 7 = 7 people 8 = 8 people 9 = 9 people	3.90	1.54	1	9
	Number of refrigerators	1 = 0 units 2 = 1 set 3 = 2 or more	0.96	0.40	1	3
	Number of air conditioners	1 = 0 units 2 = 1 set 3 = 2 sets 4 = 3 or more	1.76	0.98	1	4

**Table 2 ijerph-19-17054-t002:** Least squares (OLS) regression results.

Variables	Total Household Carbon Emissions			
	Correlation Coefficient	Standard Deviation	T Value	*p* Value
Permanent household population	0.065	0.031	2.13	0.034
Family income				
5–10 million	0.044	0.121	0.36	0.719
10–15 million	0.054	0.126	0.43	0.666
15–20 million	0.068	0.147	0.46	0.646
20–30 million	0.027	0.160	0.17	0.868
More than 300,000	0.103	0.128	0.81	0.419
Number of refrigerators				
1 unit	0.603	0.213	2.83	0.005
2 or more	0.507	0.239	2.12	0.035
Number of air conditioners				
1 unit	0.490	0.198	2.47	0.014
2 units	0.835	0.204	4.08	0.000
3 or more	0.974	0.210	4.65	0.000
Construction area				
50–70 square meters	0.049	0.126	0.39	0.698
70–100 square meters	0.283	0.146	1.94	0.053
100 square meters or more	0.282	0.150	1.87	0.062
Building age				
5–10 years	0.180	0.130	1.39	0.167
10–20 years	0.202	0.130	1.55	0.122
More than 20 years	0.264	0.152	1.74	0.083
Constant	−1.055	0.258	−4.09	0.000
Observations	247			
R-squared	0.497			

## Data Availability

Not applicable.

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
