# Peer review of "Influencing Factors of Direct Carbon Emissions of Households in Urban Villages in Guangzhou, China"

_ijerph, 2022, doi:10.3390/ijerph192417054_

Round 1
Reviewer 1 Report
This study investigated the influencing factors and direct carbon emissions of households in urban villages in Guangzhou by using micro level household survey data and a multiple regression model. This study has also done many works on exploring the energy consumption structure and end uses leading to carbon emissions. I think the paper is rather interesting, and presents the obtained data well. I have some suggestions for large items to tackle (along with some minor gripes) that should be addressed before publication in order to make the paper more useful to other practitioners in the field.
1. The abstract is not much clear. It is needs to be reorganized and improved. Give to clear the method, strategies, especially new findings of the paper.
2. Is it more accurate to replace the statement of scholars in the paper with a statement from researchers?
3. Could you please confirm that the quotation marks in the paper are formatted correctly?
4. In order to quote the author's opinion, please use the standard name expression format. In particular, lines 55 and 63 in this paper.
5. I would like you to carefully check the tenses used in this paper. This paper particularly emphasizes the lines 69 and 79.
6. The citation format in line 136 needs to be checked.
7. In 2. Calculation and analysis of household energy carbon emissions, formulas should be edited in this paper by professional editors. For example, the software of MathType should be used.
8. In this paper, figures and tables are expressed in canonical format.
9. It is important to determine whether the font size of the title in Figure 5 is consistent with other titles in this paper.
10. The results should be logically-structured, involving the main and interesting conclusions.
11. It is the author's responsibility to ensure that the reference format is correct.
12. A review by a native English speaker is required in order to make the paper more intelligible and easier to follow, also as regards the abstract.
Major issues:
(1) The Abstract is the epitome of the whole paper, which reflects the theme, purpose and the main/core findings (not all the findings) of the paper. Therefore, the Abstract needs to be improved.
(2) There is a lack of clarity in the logic and structure of the introduction. It is suggested that simplification and conciseness should be closely related to the content of each paragraph. Make sure that the overall logic is rigorous and clear. There should be a clear internal logic within each paragraph in which methodology, content, and findings can be discussed one by one.
(3) Literature reviews should be focused on the research question or purpose of the paper. It is not necessary to include non-relevant literature. Additionally, literature that is not closely related to the research content can be omitted.
(4) Methods and Results: It’s better to separate the methods from the results. Specifically, the second part of the paper should be “Methods and data” (including research area and data source); and the third part of the paper should be “Results”.
(5) Conclusion: This part is interesting, but the authors need to summarize the research from both theoretical and practical significance and carefully delineate contributions.
(6) As for the last section, in my view, it is better to mention the limitations of the study and the future research.
Minor issues:
(1) In Figure 3: What do orange bars and green bars represent?
(2) All the tables and formulas should be well arranged. For instance, try to keep one table in one page, do not separate a table into two pages without any table header or note.
(3) About the formatting in the text:
- Line 141: (Qian, et al.,2016)
- Lines 373-381
Author Response
International Journal of Environmental Research and Public Health: No. ijerph-2068672
Response to Reviewers' Comments
> Our replies are preceded with the > sign and in blue. We have also provided a version of the revised manuscript where all changes are marked (in addition to a clean version).
Reviewer #1 (Remarks to the Author):
This study investigated the influencing factors and direct carbon emissions of households in urban villages in Guangzhou by using microlevel household survey data and a multiple regression model. This study has also explored the energy consumption structure and end uses leading to carbon emissions. I think the paper is rather interesting, and presents the obtained data well. I have some suggestions for large items to tackle (along with some minor gripes) that should be addressed before publication in order to make the paper more useful to other practitioners in the field.
> Thank you very much for your constructive and helpful comments.
- The abstract is not much clear. It is needs to be reorganized and improved. Give to clear the method, strategies, especially new findings of the paper.
>Thank you very much for your suggestion.
The abstract has been revised.
China’s household energy consumption has obvious regional differences, and rising income level and urbanization have changed the ability of households to make energy consumption choices. In this paper, we analyze the energy consumption characteristics of urban village residents based on microlevel household survey data from urban villages in Guangzhou, China. Then, the results of modeling the material flows of per capita carbon emissions show the most dominant type of energy consumption. OLS is applied to analyze the influencing factors of carbon emissions. We find that the per capita household carbon emissions in urban villages are 722.7 kg/year, and the average household carbon emissions are 2820.57 kg/year. We also find that household characteristics, household size, household appliance numbers, and carbon emissions have a significant positive correlation, while income has no significant effect on carbon emissions. What is more, the size and age of the house have a positive impact on carbon emissions. Otherwise, the new finding is the demonstration that income is not significantly correlated with household carbon emissions, which is consistent with the characteristics of urban villages described earlier. On the basis of this study, we propose more specific recommendations regarding household energy carbon emissions in urban villages.
- Is it more accurate to replace the statement of scholars in the paper with a statement from researchers?
> Thank you very much for your suggestion.
The paper has been revised. In particular, the modified contents are shown in lines 33, 53, 57, 61, 105 and 406.
- Researchers are paying increasing attention to the relevance of carbon emissions and household elements, and they have proposed many macro and micro policies to achieve carbon emission reduction.
- Many researchers have considered urban villages from the perspective of urbanization and social issues.
- However, researchers have paid little attention to the energy consumption and carbon emissions of urban village households.
- Researchers have already carried out many studies in the field of household energy consumption.
- Generally, Chinese researchers’ research on carbon emissions from household energy consumption is divided into the following areas: from the perspective of research, there are studies from the perspective of direct energy consumption.
- Many researchers believe that urban villages are an obstacle to sustainable urban development.
- Could you please confirm that the quotation marks in the paper are formatted correctly?
> Thank you very much for your suggestion.
- We have revised line 128 of the paper as follows.
While India and Bangladesh have slums, urban villages are very "Chinese".
- We have revised line 135 of the paper as follows.
Urban villages are very much "Chinese in character", with rapid urbanization driving regional economic growth and leading to the emergence of informal urban living spaces.
- To quote the author's opinion, please use the standard name expression format. In particular, lines 55 and 63 in this paper.
> Thank you very much for your suggestion.
- We have revised line 55 of the paper as follows.
Gao believes that urban villages help immigrants integrate into urban society by providing affordable housing and convenient facilities.
- We have revised line 62 of the paper as follows.
Zhang found the factors influencing energy consumption and direct and indirect CO2 emissions that are related due to household consumption using the input‒output method.
- We have revised line 73 of the paper as follows.
Du used a cointegration econometric model to analyze how China's economic growth potential is affected by macroeconomic variables in the context of urbanization and to forecast China's economic growth potential up to 2020 under different scenarios.
- We have revised line 77 of the paper as follows.
Wang used an input‒output model to calculate the direct and indirect carbon emissions of urban and rural residents' consumption in the BTH region from 2002 to 2012. The results showed that the direct and indirect carbon emissions of residents' consumption increased year by year.
- We have revised line 102 of the paper as follows.
Xu found that residence area has a significant impact on household energy consumption and carbon emissions.
- We have revised line 114 of the paper as follows.
Nair studied the correlation between household carbon emissions and income and household size in three Indian cities using ANOVA and linear regression models to show that household size and CO2 emissions from different activities had a large impact on total emissions.
- We have revised line 117 of the paper as follows.
Khosla discussed the changing demand for energy services among low-income households in urban India and showed that on the energy demand side, appliance penetration rises as household consumption capacity increases.
- We have revised line 121 of the paper as follows.
Baul used a semistructured questionnaire to conduct an exploratory survey of 189 households in three income groups in the suburbs of Chittagong, Bangladesh.
- I would like you to carefully check the tenses used in this paper. This paper particularly emphasizes lines 69 and 79.
> Thank you very much for your suggestion.
The tenses used in this paper have been revised.
- The results showed that direct energy consumption and CO2 emissions were the main components of total energy consumption and CO2 emissions from household consumption, and indirect CO2 emissions showed an increasing trend.
- The results of the study showed that lifestyle, standard of living and energy prices had significant effects on CO2
- The citation format in line 136 needs to be checked.
> Thank you very much for your suggestion.
The citation format in line 136 has been revised.
- In 2. Calculation and analysis of household energy carbon emissions,formulas should be edited in this paper by professional editors. For example, MathType software should be used.
> Thank you very much for your suggestion.
The quotations have been modified.
(1)
(2)
(3)
(4)
(5)
- In this paper, figures and tables are expressed in canonical format.
> Thank you very much for your valuable comments.
The figures and tables in the paper have been reviewed by me.
- It is important to determine whether the font size of the title in Figure 5 is consistent with other titles in this paper.
> Thank you very much for your suggestion.
Figure 5 has been revised.
Figure 5. Modeling of material flows of carbon emissions per capita from household energy consumption.
- The results should be logically structured, involving the main and interesting conclusions.
> Thank you very much for your valuable comments.
The result has been revised.
In terms of the theoretical contribution, the conclusion of the paper is conducive to reducing carbon emissions, promoting the construction and development of low-carbon communities, and mitigating global warming. It contributes to the promotion of energy efficiency and emission reduction awareness among consumers, as well as reducing carbon emissions from the consumption side of the equation. What is more, the practical contribution consists of facilitating residents' participation in the development of low-carbon, sustainable communities. In addition, it is beneficial for building the sustainable community, especially for the lower income households. Thus, the results of our study can provide policymakers with a good case to make policy.
- It is the author's responsibility to ensure that the reference format is correct.
> Thank you very much for your suggestion. All references have been checked.
- A review by a native English speaker is required in order to make the paper more intelligible and easier to follow, also as regards the abstract.
> Many thanks for your suggestion. The native speakers have made corrections.
Major issues:
- The Abstract is the epitome of the whole paper, which reflects the theme, purpose and the main/core findings (not all the findings) of the paper. Therefore, the Abstract needs to be improved.
>Thank you very much for your suggestion.
The abstract has been revised.
China’s household energy consumption has obvious regional differences, and rising income level and urbanization have changed the ability of households to make energy consumption choices. In this paper, we analyze the energy consumption characteristics of urban village residents based on microlevel household survey data from urban villages in Guangzhou, China. Then, the results of modeling the material flows of per capita carbon emissions show the most dominant type of energy consumption. OLS is applied to analyze the influencing factors of carbon emissions. We find that the per capita household carbon emissions in urban villages are 722.7 kg/year, and the average household carbon emissions are 2820.57 kg/year. We also find that household characteristics, household size, household appliance numbers, and carbon emissions have a significant positive correlation, while income has no significant effect on carbon emissions. What is more, the size and age of the house have a positive impact on carbon emissions. Otherwise, the new finding is the demonstration that income is not significantly correlated with household carbon emissions, which is consistent with the characteristics of urban villages described earlier. On the basis of this study, we propose more specific recommendations regarding household energy carbon emissions in urban villages.
- There is a lack of clarity in the logic and structure of the introduction. It is suggested that simplification and conciseness should be closely related to the content of each paragraph. Make sure that the overall logic is rigorous and clear. There should be a clear internal logic within each paragraph in which methodology, content, and findings can be discussed one by one.
>Thank you very much for your suggestion.
The logic in the introduction has been revised. The following revision has been made to line 40 of the paper.
Reducing carbon emissions and promoting the construction of low-carbon communities are important ways to promote the development of low-carbon communities and mitigate global warming. As a result of the literature, it has been difficult to identify a model from the social development of other countries that can be used for reference. The purpose of the paper is to supplement the theoretical deficiencies of these studies. This paper focuses on the energy consumption behavior, characteristics, and energy saving potential of this group of people.
- Literature reviews should be focused on the research question or purpose of the paper. It is not necessary to include nonrelevant literature. Additionally, literature that is not closely related to the research content can be omitted.
>Thank you very much for your suggestion.
The literature reviews have been revised. We have reviewed the irrelevant literature. The following revision has been made to line 105 of the paper.
In general, researchers who study carbon emissions from household energy consumption fall into the following categories. In terms of research, there is direct and indirect energy consumption. For area characteristics, there are urban and rural areas. For data use, both survey data and statistical data are available. In some studies, the influencing factors are examined. It is primarily influenced by socioeconomic and geographical factors. In previous studies, household income, age, household size, education level, location, gender, and building time have been shown to have the greatest impact on carbon emissions. Nevertheless, few researchers have examined household carbon emissions from the perspective of urban communities.
(4) Methods and Results: It is better to separate the methods from the results. Specifically, the second part of the paper should be “Methods and data” (including research area and data source); and the third part of the paper should be “Results”.
>Thank you very much for your suggestion.
We have revised the serial number as follows.
- Data and methodology
2.1. Research area
2.2. Data sources and sample characteristics
2.3. Calculation and analysis of household energy carbon emissions
2.3.1. Household carbon emissions calculation
2.3.2. Basic information on household carbon emissions
2.3.3. Comparative analysis with other cities in China
2.4. Modeling of carbon emissions energy flows per capita
- Results
3.1. Variable selection and description
3.2. OLS regression model
3.3. Regression analysis
(5)Conclusion: This part is interesting, but the authors need to summarize the research from both theoretical and practical significance and carefully delineate contributions.
>Thank you very much for your suggestion.
- Question 1:
We have revised line 395 theoretical and practical significance as follows:
In terms of the theoretical contribution, the conclusion of the paper is conducive to reducing carbon emissions, promoting the construction and development of low-carbon communities, and mitigating global warming. It contributes to the promotion of energy efficiency and emission reduction awareness among consumers, as well as reducing carbon emissions from the consumption side of the equation. What is more, the practical contribution consists of facilitating residents' participation in the development of low-carbon, sustainable communities. In addition, it is beneficial for building the sustainable community, especially for the lower income households. Thus, the results of our study can provide policymakers with a good case to make policy.
- Question 2:
The contributions are specified in the discussion. We have revised line 360 as follows.
To explain the reasons behind these phenomena, we modeled the factors that influence household energy consumption in urban villages in Guangzhou. Comparatively, existing studies have focused more on the temporal variation and urban‒rural differences in household carbon emissions from energy consumption [38]. In this study, we examine a unique community property, the urban village, from the perspective of a new perspective. There is no significant effect of household income on carbon emissions in urban villages in Guangzhou, which is inconsistent with other research. We have also examined the impact of building time and floor area on carbon emissions in urban villages in the hope of providing some suggestions for creating green, low-carbon and energy-efficient buildings. Our findings provide insight into the direct carbon emissions associated with particular communities and can serve as a basis for future research [39].
(1) The characteristics of households
There is no significant correlation between household income and carbon emissions. The findings are inconsistent with those of Meangbua and De [40,41]. According to these two researchers, household income has a significant impact on carbon emis-sions from energy consumption. In spite of this, our study indicates that there is no significant correlation between household income and household energy carbon emis-sions. We believe that this is one of the most important contributions of our study since it fully explains the residential characteristics of urban villages in Guangzhou, most of whom are low-income migrant workers. The greater the number of household members, the greater the total household carbon emissions, a finding that is consistent with those of other researchers. In view of the fact that people are the primary con-sumers of energy, this is reasonable. Carbon emissions are also influenced by the number of household appliances owned [42]. Based on Tran and Lee's argument, it can be shown that the more household appliances used in a household, the greater the impact on carbon emissions [43,44].
(2) The characteristics of housing
The percentage of buildings in urban villages that are older than 15 years is 52.5%, more than half. A longer building time indicates poor thermal performance, which results in higher electricity consumption. This is consistent with the findings of Yu and Gu's study [45,16]. There is a significant impact of floor space on carbon emissions in terms of area. It is generally true that the larger the size of a household building is, the more energy is consumed for cooling. Thus, reasonable and effective control of housing size is also an effective method of achieving low carbon development [46].
(6)As for the last section, in my view, it is better to mention the limitations of the study and the future research.
>Thank you very much for your suggestion.
We have revised line 461 of the text as follows.
The study primarily relies on microsurveys for its data, whose quality and quantity are highly relevant to its credibility. We need to open up access to more accessible and relevant data to be able to study carbon emissions from household energy consumption more accurately and in depth. Due to limited resources, our study, like many survey studies, requires a larger valid sample to be accurate. In the future, follow-up studies should expand the sample size and examine the management model and sustainable development of urban village communities.
Minor issues:
- In Figure 3: What do orange bars and green bars represent?
>Thank you very much for your suggestion.
Figure 3 has been revised in the paper.
Figure 3. Analysis of carbon emissions from energy consumption by type.
(2) All the tables and formulas should be well arranged. For instance, try to keep one table in one page, do not separate a table into two pages without any table header or note.
>Thank you very much for your suggestion.
Each table in the paper has been carefully reviewed for format.
(3) About the formatting in the text: Line 141: (Qian, et al.,2016); Lines 373-381
>Thank you very much for your suggestion.
The citation format in line 141 has been revised.
The references in lines 373-381 have been revised.
Reviewer 2 Report
I am pleased to review the manuscript entitled “Influencing Factors of Direct Carbon Emissions of Households in Urban Villages in Guangzhou, China”. This study analyzes the energy consumption characteristics of urban village residents based on micro level 9 household survey data from urban villages in Guangzhou, China, and construct an energy flow 10 model of per capita carbon emissions. I have some issues to discuss with the authors.
1. In the practical background, the policy regarding “two carbon” should be provided. Additionally, author should clarify the importance of the reduction of household carbon emissions in Introduction.
2. In my opinion, I believe that the theoretical contribution needs to be given in the Introduction. I do not know what the literature contribution is?
3. As far as I am concerned, I believe that author need to write an individual literature review.
4. The serial number of “Research area and data source” is wrong. The serial number of “Introduction” is 1. Moreover, I suggest that the Section 2 and Section 3 should be integrated.
5. I strongly suggest that author should write the discussion to analyze the research results by comparing to the previous literature. Moreover, the literature contribution needs to be stressed again.
6. I think that the language need be polished by the native English speakers. In addition, the distinguishability of all figures needs to be improved.
Author Response
International Journal of Environmental Research and Public Health: No. ijerph-2068672
Response to Reviewers' Comments
> Our replies are preceded with the > sign and in blue. We have also provided a version of the revised manuscript where all changes are marked (in addition to a clean version).
Reviewer #2 (Remarks to the Author):
I am pleased to review the manuscript entitled “Influencing Factors of Direct Carbon Emissions of Households in Urban Villages in Guangzhou, China”. This study analyzes the energy consumption characteristics of urban village residents based on micro level 9 household survey data from urban villages in Guangzhou, China, and constructs an energy flow 10 model of per capita carbon emissions. I have some issues to discuss with the authors.
> We thank you for these supportive words and your constructive comments.
- In the practical background, the policy regarding “two carbon” should be provided. Additionally, author should clarify the importance of the reduction of household carbon emissions in Introduction.
> Thank you very much for your suggestion.
- Question 1:
The following revision has been made to line 31 of the paper.
China formally proposed that carbon peaking be achieved by 2030 and carbon neutrality by 2060 at the 75th United Nations General Assembly. "Double carbon" is proposed as a goal.
- Question 2:
The following revision has been made to line 40 of the paper.
Reducing carbon emissions and promoting the construction of low-carbon communities are important ways to promote the development of low-carbon communities and mitigate global warming.
- In my opinion, I believe that the theoretical contribution needs to be given in the Introduction. I do not know what the literature contribution is?
> We appreciate your suggestion very much.
The following revision has been made to line 42 of the paper.
As a result of the literature, it has been difficult to identify a model from the social development of other countries that can be used for reference. The purpose of the paper is to supplement the theoretical deficiencies of these studies. This paper focuses on the energy consumption behavior, characteristics, and energy-saving potential of this group of people.
We have added the theoretical contribution in the conclusion as follow:
In terms of the theoretical contribution, the conclusion of the paper is conducive to reducing carbon emissions, promoting the construction and development of low-carbon communities, and mitigating global warming. It contributes to the promotion of energy efficiency and emission reduction awareness among consumers, as well as reducing carbon emissions from the consumption side of the equation. What is more, the practical contribution consists of facilitating residents' participation in the development of low-carbon, sustainable communities. In addition, it is beneficial for building the sustainable community, especially for the lower income households. Thus, the results of our study can provide policymakers with a good case to make policy.
- As far as I am concerned, I believe that author need to write an individual literature review.
> My sincere thanks for your suggestion.
The paper has been revised as follows at line 105.
In general, researchers who study carbon emissions from household energy consumption fall into the following categories. In terms of research, there is direct and indirect energy consumption. For area characteristics, there are urban and rural areas. For data use, both survey data and statistical data are available. In some studies, the influencing factors are examined. It is primarily influenced by socioeconomic and geographical factors. In previous studies, household income, age, household size, education level, location, gender, and building time have been shown to have the greatest impact on carbon emissions. Nevertheless, few researchers have examined household carbon emissions from the perspective of urban communities.
- The serial number of “Research area and data source” is wrong. The serial number of “Introduction” is 1. Moreover, I suggest that the Section 2 and Section 3 should be integrated.
> My sincere thanks go out to you for your suggestion.
We have revised the serial number as follows.
- Data and methodology
- Research area
- Data sources and sample characteristics
- Calculation and analysis of household energy carbon emissions
- Household carbon emissions calculation
- Basic information on household carbon emissions
- Comparative analysis with other cities in China
2.4. Modeling of carbon emissions energy flows per capita
- I strongly suggest that author should write the discussion to analyze the research results by comparing to the previous literature. Moreover, the literature contribution needs to be stressed again.
> Thank you very much for your suggestion.
The discussion has been written as follows.
To explain the reasons behind these phenomena, we modeled the factors that influence household energy consumption in urban villages in Guangzhou. Comparatively, existing studies have focused more on the temporal variation and urban‒rural differences in household carbon emissions from energy consumption [38]. In this study, we examine a unique community property, the urban village, from the perspective of a new perspective. There is no significant effect of household income on carbon emissions in urban villages in Guangzhou, which is inconsistent with other research. We have also examined the impact of building time and floor area on carbon emissions in urban villages in the hope of providing some suggestions for creating green, low-carbon and energy-efficient buildings. Our findings provide insight into the direct carbon emissions associated with particular communities and can serve as a basis for future research [39].
(1) The characteristics of households
There is no significant correlation between household income and carbon emissions. The findings are inconsistent with those of Meangbua and De [40,41]. According to these two researchers, household income has a significant impact on carbon emissions from energy consumption. In spite of this, our study indicates that there is no significant correlation between household income and household energy carbon emissions. We believe that this is one of the most important contributions of our study since it fully explains the residential characteristics of urban villages in Guangzhou, most of whom are low-income migrant workers. The greater the number of household members, the greater the total household carbon emissions, a finding that is consistent with those of other researchers. In view of the fact that people are the primary consumers of energy, this is reasonable. Carbon emissions are also influenced by the number of household appliances owned [42]. Based on Tran and Lee's argument, it can be shown that the more household appliances used in a household, the greater the impact on carbon emissions [43,44].
(2) The characteristics of housing
The percentage of buildings in urban villages that are older than 15 years is 52.5%, more than half. A longer building time indicates poor thermal performance, which results in higher electricity consumption. This is consistent with the findings of Yu and Gu's study [45,16]. There is a significant impact of floor space on carbon emissions in terms of area. It is generally true that the larger the size of a household building is, the more energy is consumed for cooling. Thus, reasonable and effective control of housing size is also an effective method of achieving low carbon development [46].
- I think that the language need be polished by the native English speakers. In addition, the distinguishability of all figures needs to be improved.
> Many thanks for your suggestion.
- Question 1:
The native speakers have made corrections.
- Question 2:
We have made adjustments to all figures.
Reviewer 3 Report
Please find attached my comments
Best regards

Author Response
I am pleased to present to you my opinion after reviewing this research (case study). The submission topic is attractive and well done, the article contains good aspects, the article is written in an acceptable manner that reflects the authors' understanding of the topic, and recent references relevant to the research point are presented in a sufficiently integrated and up-to-date manner.
The particle size is appropriate for the content the text is presented in a smooth manner that scholars in other disciplines can easily understand and the text is clearly and concisely arranged.
The search summary adequately covers the article's contents and suitable keywords by which the article can be easily found when searching for it in records or indexes, and the conclusions are well explained, justified and understandable. Finally, the title of the plaster expresses the contents of the paper well enough.
There are some minor modifications that must be made to make the article look better before publishing, such as:
> We thank you for these supportive words and your constructive comments.
- Add a list of abbreviations at the end of the article
>Thank you very much for your suggestion.
We have added a list of abbreviations at the end of the article.
|
Abbreviation |
|
|
LPG |
Liquefied Petroleum Gas |
|
CO2 emissions |
Carbon emissions |
|
TV |
Television |
- In most parts of the research, the sentences are very long and cannot be understood in this way - the sentences must be short and clear for ease of understanding for the reader
> Many thanks for your suggestion. The native speakers have made corrections.
- 5. Conclusions and recommendations - Delete letter(s) from Conclusion
>Thank you very much for your suggestion.
We have revised 5. Conclusions and recommendations as follows.
Conclusion and recommendations
- It is preferable to write the summary in a table for ease of understanding
>Thank you very much for your suggestion. Considering the length of the paper, we haven't added the table for the time being, but we will consider your suggestion in future research